# Multiplex Soluble Biomarker Analysis from Pleural Effusion

**DOI:** 10.3390/biom10081113

**Published:** 2020-07-28

**Authors:** Joman Javadi, Katalin Dobra, Anders Hjerpe

**Affiliations:** 1Karolinska Institutet, Department of Laboratory Medicine, Division of Pathology, Huddinge University Hospital, SE-14186 Stockholm, Sweden; katalin.dobra@ki.se; 2Karolinska University Hospital, Karolinska University laboratory, Huddinge University Hospital, SE-14186 Stockholm, Sweden; anders.hjerpe@sll.se

**Keywords:** malignant pleural mesothelioma, pleural effusion, Luminex, diagnostic biomarkers

## Abstract

Malignant pleural mesothelioma (MPM) is a highly aggressive and therapy resistant pleural malignancy that is caused by asbestos exposure. MPM is associated with poor prognosis and a short patient survival. The survival time is strongly influenced by the subtype of the tumor. Dyspnea and accumulation of pleural effusion in the pleural cavity are common symptoms of MPM. The diagnostic distinction from other malignancies and reactive conditions is done using histopathology or cytopathology, always supported by immunohistochemistry, and sometimes also by analyses of soluble biomarkers in effusion supernatant. We evaluated the soluble angiogenesis related molecules as possible prognostic and diagnostic biomarkers for MPM by Luminex multiplex assay. Pleural effusion from 42 patients with malignant pleural mesothelioma (MPM), 36 patients with adenocarcinoma (AD) and 40 benign (BE) effusions were analyzed for 10 different analytes that, in previous studies, were associated with angiogenesis, consisting of Angiopoietin-1, HGF, MMP-7, Osteopontin, TIMP-1, Galectin, Mesothelin, NRG1-b1, Syndecan-1 (SDC-1) and VEGF by a Human Premixed Multi-Analyte Luminex kit. We found that shed SDC-1 and MMP-7 levels were significantly lower, whereas Mesothelin and Galectin-1 levels were significantly higher in malignant mesothelioma effusions, compared to adenocarcinoma. Galectin-1, HGF, Mesothelin, MMP-7, Osteopontin, shed SDC-1, NRG1-β1, VEGF and TIMP-1 were significantly higher in malignant pleural mesothelioma effusions compared to benign samples. Moreover, there is a negative correlation between Mesothelin and shed SDC-1 and positive correlation between VEGF, Angiopoietin-1 and shed SDC-1 level in the pleural effusion from malignant cases. Shed SDC-1 and VEGF have a prognostic value in malignant mesothelioma patients. Collectively, our data suggest that MMP-7, shed SDC-1, Mesothelin and Galectin-1 can be diagnostic and VEGF and SDC-1 prognostic markers in MPM patients. Additionally, Galectin-1, HGF, Mesothelin, MMP-7, Osteopontin, shed SDC-1 and TIMP-1 can be diagnostic for malignant cases.

## 1. Introduction

Malignant pleural mesothelioma (MPM) is an aggressive tumor of mesothelial origin that occurs after a long latency period following asbestos exposure often comprising several decades [1,2,3,4]. This tumor occurs mostly in the pleura (>75%) and less often in the peritoneum (10–20%) or pericardium (1%) [5,6]. The median survival time of malignant pleural mesothelioma is less than 12 months [7]. This survival is strongly influenced by subtype of the tumor and time of diagnosis [3,8]. The diagnosis of MPM is challenging, which often leads to a late diagnosis and together with tumor aggressiveness results in a poor prognosis. Diagnosis is based on testing of multiple immunohistochemical (IHC) markers of biopsies or cytological samples, and soluble biomarker analysis of pleural effusion to distinguish MPM from other lung malignancies [9]. The optimal treatment is surgery in combination with chemotherapy and radiotherapy, or both [10]. Most patients present with a pleural effusion as first symptom and cytological diagnosis increases the chance for earlier diagnosis and prolongs patient survival [11]. Among soluble biomarkers hyaluronan and mesothelin are the best characterized biomarkers for diagnosis of MPM.

Hyaluronan (HA), a negatively charged glycosaminoglycan of the extracellular matrix (ECM), consists of the repeating disaccharide units (glucuronic acid and *N*-acetylglucosamine), is an established biomarker used in diagnosis of MPM [12,13,14]. Hyaluronan is involved in different functions such as cell growth, angiogenesis, differentiation, cell migration, wound healing and the regulation of plasma protein distribution [13,15]. Hyaluronan is a well-known biomarker for MPM [14,16,17,18].

Mesothelin is a 40-KD cell membrane glycoprotein, which presents on normal mesothelial cells and is highly expressed in different human cancers of the lung, endometrium, stomach and pancreas [19]. Mesothelin is synthesized as a 72-KD protein anchored in the cell membrane. It sheds a 32-KD fragment called *N*-ERC/mesothelin, leaving a 40-KD fragment named C-ERC/mesothelin. Both fragments can be detected in the effusion supernatant [20]. The role of mesothelin in the development of normal cells is not known, whereas, in cancer cells, mesothelin promotes tumor progression by interacting with other membrane proteins, such as CA125 [21,22,23]. In recent years, Mesothelin has received considerable attention for immune-based therapies and as a promising biomarker, because of its low expression level in normal cells [24,25].

Syndecan-1 (SDC-1) is a transmembrane heparan sulfate proteoglycan containing glycosaminoglycan (GAG) side chains linked to *N*-terminal extracellular domain of the core protein. Syndecan-1 acts as a co-receptor for many regulatory proteins, such as growth factors, cytokines and integrins through its GAG, mainly heparan sulfate, chains. This interaction leads to regulation of different cellular processes including proliferation, migration, differentiation and angiogenesis by alteration of different signaling pathways [26,27]. In different cancer types syndecan-1 plays dynamic roles, whereby it can either suppress or promote tumor progression [28,29,30,31].

The aim of this study is to evaluate shed SDC-1 in pleural effusions together with angiogenesis related proteins and identify optimal biomarker batteries that allow earlier diagnosis and improve the possibilities to distinguish MPM from metastatic adenocarcinoma and reactive conditions.

## 2. Materials and Methods

### 2.1. Study Design

Pleural effusions from 118 patients—42 patients with malignant pleural mesothelioma (MPM), 36 patients with lung adenocarcinoma (AD) and 40 benign (BE) effusions—were collected at the Department of Pathology and Cytology, Karolinska University Hospital Sweden. Malignant mesothelioma effusions comprised the epithelioid and mixed phenotypes, as sarcomatoid mesotheliomas do not exfoliate to the serous effusions. Adenocarcinoma specimens consisted of metastases from lung, breast- gastro-intestinal, ovarian adenocarcinoma and primary tumors of unknown primary, covering the most frequent metastatic tumors to the serosal cavities. Benign effusions were related to inflammation, reactive mesothelial proliferations and heart failure. All cases were diagnosed by cytopathology verified by extensive immunocytochemistry. Mesothelioma cases were also analyzed by established biomarker analyses including Mesothelin and Hyaluronan. All samples were collected before any treatment was given. Samples were centrifuged at 1500 rpm for 5 min directly, and the cell free supernatants were kept at −80 °C without additives. The study was approved by the ethical review board of Stockholm, Sweden (2009/1138–341/3).

### 2.2. Luminex Assay with Human Premixed Multi-Analyte Kit

Two human premixed multi-analyte kits from the R&D system were used to assess the levels of 10 different biomarkers. First kit (cat: LXSAHM-09) for analyzing Angiopoietin-1, HGF, MMP-7, Osteopontin, TIMP-1, Galectin, Mesothelin, NRG1-b1 and Syndecan-1 simultaneously and second kit (cat: LXSAHM-01) for analyzing VEGF. In total, we analyzed 118 pleural effusions, of which 42 were from MM patients, 36 from AD patients and 40 benign effusions. Effusions were diluted 5-fold, using the dilution buffer included in the kit. All standards and samples were assayed in duplicate.

Analyte specific antibodies are pre-coated onto magnetic microparticles embedded with fluorophores at set ratios for each unique microparticle region. Two spectrally distinct light emitting diodes (LEDs) illuminate the microparticles. One LED excites the dyes inside each microparticle, to identify the region and the second LED excites the Streptavidin-PE to measure the amount of analyte bound to the microparticles. A minimum of 50 beads per region were counted. The median fluorescence intensities were determined on a Luminex^®^ 100/200^™^ analyzer.

### 2.3. Enzyme-Linked Immunosorbent Assay (ELISA)

Enzyme-linked immunosorbent assay (ELISA) was performed to measure shed SDC-1, VEGF and Mesothelin levels, following the manufacturer’s instructions. ELISA kit for Human shed SDC-1 was from Gen-Probe Diaclone, France (cat. number 950.640.192), Human VEGF was from R&D Systems, UK (cat. number DVE00) and Human *N*-ERC/Mesothelin Assay kit from IBL. The effusions were diluted 1:5 in kit dilution buffers which also were as blanks. Samples were analyzed in duplicate and the optical densities were determined at 450 nm.

### 2.4. Statistical Analyses

#### 2.4.1. Analyses of Biomarkers Expression in Pleural Effusions 

Three different groupings of patients were used in the analysis of ten different biomarkers, to compare biomarkers expression level between patients with malignant and benign conditions, and also to compare patients with different malignancies comprising malignant mesothelioma (MM) and various adenocarcinomas (AD). Differences in biomarkers expression level between different patient groups were evaluated by performing an unpaired t-test. Correlation analysis between shed SDC-1 and other biomarkers in paired pleural effusions were calculated by the Spearman r test.

#### 2.4.2. Kaplan-Meier Survival 

Analyses were performed for patients with malignant mesothelioma, to investigate the correlation of different biomarkers level with the survival of mesothelioma patients. To determine a cut-off value for each biomarker based on the most significant and highest hazard ratio we used the Cutoff Finder online web application. The log-rank test was used to test the significant differences between survival times of two different groups of patients.

#### 2.4.3. Receiver Operating Characteristic (ROC) Analysis 

Roc analysis was used to assess the diagnostic utility of each biomarker. ROC curves, areas under the curve (AUC), sensitivity, specificity, likelihood ratio and their 95% confidence intervals (CI) were generated by GraphPad Prism software. All statistical significances were set at a *p* value equal or lower than 0.05.

#### 2.4.4. Logistic Regression (LR) Analysis 

Logistic regression analysis was applied to develop a model for identifying and combine diagnostic biomarkers in pleural effusion for earlier detection of malignant pleural mesothelioma. Angiogenesis-related biomarkers were analyzed using the JMP program. In each model, malignant mesothelioma (MM) and metastatic adenocarcinoma (AD) were set as dependent variable (coded 0 or 1), and the expression values of the angiogenesis-related biomarkers as independent variables. Biomarkers included in a final predictive model were significant at p≤ 0.05 and were determined by a stepwise selection procedure.

## 3. Results

### 3.1. Correlation between ELISA and Luminex Immunoassays

In order to evaluate the correlation between ELISA and Luminex immunoassays, pleural effusion levels of shed SDC-1, Mesothelin and VEGF were measured independently by both immunoassays. High correlation between immunoassays was observed (Figure 1). We found statistically significant linear correlation between shed SDC-1 (*p* = 0.0001, *r* = 0.8), Mesothelin (*p* = 0.0001, *r* = 0.6) and VEGF (*p* = 0.0001, *r* = 0.9) that measured by ELISA comparing with the Luminex assay, suggesting that Luminex assay can be used in the clinic.

### 3.2. Expression Levels of Biomarkers in Malignant Pleural Mesothelioma and Benign Pleural Effusion

In order to describe the diagnostic power of pleural effusion derived Angiopoietin-1, HGF, MMP-7, Osteopontin, TIMP-1, Galectin, Mesothelin, NRG1-b1, Syndecan-1 and VEGF, we compared the expression level of these 10 angiogenesis related biomarkers in pleural effusion from malignant pleural mesothelioma patients (*n* = 42) and benign samples (*n* = 40). The expression levels of Galectin-1, Mesothelin, Osteopontin, shed SDC-1, VEGF, MMP-7, HGF and TIMP-1 were significantly higher, and NRG1-β1 was significantly lower in malignant mesothelioma effusions, compared with the benign effusions (Figure 2). Expression level of these 10 angiogenesis related biomarkers in pleural effusion from metastatic lung adenocarcinoma patients (*n* = 38) showed in Appendix A. Differences between malignant mesothelioma cases and benign cases were statistically significant for these nine biomarkers (*p* < 0.05).

### 3.3. Diagnostic Biomarkers for Distinguishing Malignant Mesothelioma from Metastatic Adenocarcinoma

In order to distinguishing malignant mesothelioma from other adenocarcinoma, the expression levels of these 10 angiogenic related biomarkers in pleural effusion were compared between malignant pleural mesothelioma patients and metastatic adenocarcinomas. Of these, expression level of Mesothelin and Galectin-1 were significantly higher in malignant mesothelioma effusions, compared to adenocarcinoma effusions whereas the expression level of shed Syndecan-1 and MMP-7 were significantly lower in malignant mesothelioma effusions compared to the adenocarcinoma effusions (Figure 3). Differences between malignant mesothelioma and adenocarcinoma were statistically significant for these four biomarkers (*p* < 0.05).

Stepwise logistic regression based on biomarker levels showed that MMP-7, Mesothelin and Osteopontin are three variables with a higher predictive value for distinguishing malignant mesothelioma from metastatic adenocarcinoma (Table 1).

### 3.4. Shed SDC-1 and VEGF Levels Correlate to Patient Survival in Malignant Mesothelioma

In order to determine the prognostic value of pleural effusion derived Angiopoietin-1, HGF, MMP-7, Osteopontin, TIMP-1, Galectin, Mesothelin, NRG1-b1, SDC-1 and VEGF, we divided patients in two groups depending on the established cut-off value for all analytes. Strikingly, malignant mesothelioma patients with high levels of SDC-1 (>19.28 ng/mL) and VEGF (>0.7 ng/mL) have significantly worse prognosis in comparison to the patients with low levels of SDC-1 (<19.28 ng/mL) and VEGF (<0.7 ng/mL). Median survival time of malignant mesothelioma patients with high VEGF level was significantly shorter (2.6 months) compared to low VEGF level (18 months) (*p* = 0.0003) (Figure 4A). Median survival time of malignant mesothelioma patients with high level of SDC-1 was significantly shorter (2.4 months) compared to patients with low shed SDC-1 level (9.6 months) (*p* = 0.03) (Figure 4B).

### 3.5. Correlation between Shed SDC-1 and Other Biomarkers

The correlation between shed SDC-1 and other biomarkers was explored in malignant mesothelioma patients and in all malignant cases (malignant mesothelioma and adenocarcinoma). In all malignant cases, we showed a significant weak-positive correlation between shed SDC-1 and Angiopoietin-1 (*p* = 0.004 and r = 0.3) and a significant weak-negative correlation between shed SDC-1 and Mesothelin (*p* = 0.004 and r = −0.3) (Figure 5). We did not find any significant correlation between expression of shed SDC-1 and other seven angiogenic-related biomarkers (HGF, MMP-7, Osteopontin, TIMP-1, Galectin, VEGF and NRG1-b1).

In malignant mesothelioma cases, shed SDC-1 expression was significantly and positively correlated with HGF (*p* = 0.02; r = 0.3) and NRG1-b1 (*p* = 0.001; r = 0.4) (Figure 6). No significant correlations were found between shed SDC-1 and other seven biomarkers (Angiopoietin-1, Mesothelin, MMP-7, Galectin-1, Osteopontin, TIMP-1 and VEGF).

### 3.6. Diagnostic Value of Individual Biomarkers for Malignant Mesothelioma

Based on ROC curve analyses, Galectin-1, Mesothelin, Osteopontin, NRG1-β1 and shed SDC-1 performed the best diagnostic capacity to distinguish malignant mesothelioma from benign effusions. MMP-7 and angiopoietin-1 have no discriminative capacity to distinguish between malignant mesothelioma and benign effusions (Figure 7). The resulting areas under the curves (AUC) are shown in (Figure 7).

## 4. Discussion

Malignant mesothelioma is the locally very aggressive and incurable primary tumor of serous surfaces. Several studies have indicated that clinical outcome of malignant mesothelioma is highly affected by earlier diagnosis of the disease [11,32,33]. Diagnosis of malignant mesothelioma is challenging and requires histological or cytological analysis which can be supported by the analysis of soluble biomarkers in the effusion. Pleural effusion is the first available material for diagnosis, and a possible source for biomarker analysis. Previous studies showed that, Calretinin, Wilms tumor protein 1 (WT-1), HBME-1, D2-40 (podoplanin), Carcinoembryonic antigen (CEA), Napsin-A and Thyroid transcription factor 1 (TTF-1) are immunocytochemical indicators for distinguishing malignant mesothelioma from lung adenocarcinoma [3,34,35]. Calretinin, WT-1 and D2-40 are recommended mesothelioma markers and CEA, Napsin-A and TTF-1 are recommended adenocarcinoma markers by IMIG (International Mesothelioma Interest Group) [36].

In addition, soluble biomarkers are detectable, and can be easily quantified in body fluids before clinical symptoms appear. Several diagnostic biomarkers have been suggested for malignant mesothelioma. Mesothelin, Hyaluronan, Osteopontin and Fibulin-3 are the most promising diagnostic biomarker candidates for malignant mesothelioma [12,37,38,39,40]. In this regard hereby we combine multiplex soluble diagnostic and prognostic biomarkers in pleural effusion that can help to establish earlier diagnosis of malignant mesothelioma and to distinguish it from metastatic adenocarcinomas.

To the best of our knowledge, this is the first study that simultaneously combines several soluble diagnostic biomarkers. Here we show that Malignant mesothelioma patients have significantly higher level of Galectin-1, Mesothelin, Osteopontin, VEGF, shed SDC-1, MMP-7, HGF, NRG1-β1 and TIMP-1 compared to benign patients. Pleural effusion Galectin-1, NRG1-β1, Osteopontin, Mesothelin, shed SDC-1, VEGF and TIMP-1 levels are more reliable diagnostic biomarkers than HGF and MMP in pleural effusion. The corresponding AUCs were 0.99, 0.97, 0.95, 0.94, 0.93, 0.84 and 0.83 for Galectin-1, NRG1-β1, Osteopontin, Mesothelin, shed SDC-1, VEGF and TIMP-1, respectively, whereas the AUCs were 0.75 and 0.56 for HGF and MMP-7, respectively. Higher Mesothelin, Osteopontin and VEGF levels were described earlier as individual markers in malignant mesothelioma [41,42,43]. A recent study showed also higher level of TGF-β l in malignant pleural mesothelioma patients compared to lung adenocarcinoma patients [42]. Previously, we have shown that overexpression of membrane-bound SDC-1 down-regulate TGF-β and TGF-βR1 [44] and the interplay between these two components merits further investigations. High level of shed SDC-1 associates to cancer, infection and inflammation, thus our findings are in line with both previous and recent studies.

In this study, all patients with metastatic adenocarcinoma had significantly higher level of Galectin-1, HGF, Mesothelin, MMP-7, Osteopontin, shed SDC-1, TIMP-1, VEGF and a significantly lower level of NRG1-β1 compared with reactive effusions (Appendix A). Among these angiogenesis related biomarkers, we demonstrated that Galectin-1, Mesothelin, shed SDC-1 and MMP-7 are biomarkers that discriminated best between malignant mesothelioma and metastatic adenocarcinomas. Our data shows that the level of Galectin-1 and mesothelin are significantly higher in MM patients in comparison with metastatic adenocarcinoma patients whereas, shed SDC-1 and MMP-7 levels are significantly decreased in malignant mesothelioma patients. We further combined these biomarkers in several models by using logistic regression method. In our study, a combination of MMP-7, Mesothelin and Osteopontin, showed the best significant model for distinguishing malignant pleural mesothelioma from metastatic adenocarcinoma patients.

Cell surface syndecan-1 expression is essential for the differentiation of various epithelial tumors and it correlates with favorable outcome, whereas decrease or loss of SDC-1 associates with poor survival [45,46,47,48]. The elevated SDC-1 level is a result of accelerated shedding or cell decay. Here we show that the shed SDC-1 in contrast to the cell-bound SDC-1 indicates poor prognosis in both malignant pleural mesothelioma and metastatic adenocarcinoma patients. Our results clearly suggest that SDC-1 and VEGF can serve as prognostic biomarkers for malignant pleural mesothelioma. Moreover, inclusion of these soluble factors in the clinical workflow may pave the way for biomarker driven patient selection for antiangiogenic therapy in the future. Though these results are very promising, further studies with larger sample sizes are required to validate these data.

## 5. Conclusions

Taken together, candidate biomarkers (Galectin-1, Mesothelin, Osteopontin, shed SDC-1, VEGF, MMP-7, HGF, TIMP-1 and NRG1-β1) identified in the current study could be diagnostic biomarkers for distinguishing malignant pleural mesothelioma and metastatic adenocarcinoma from benign patients. Shed SDC-1 and VEGF are prognostic biomarkers for malignant pleural mesothelioma.

## Figures and Tables

**Figure 1 biomolecules-10-01113-f001:**
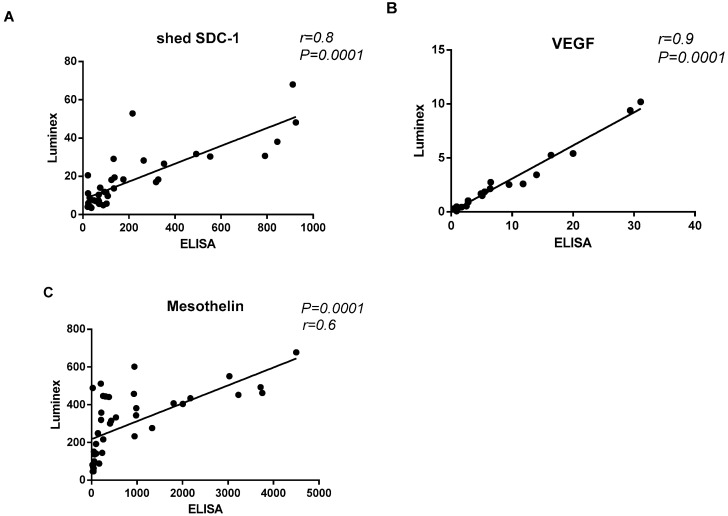
Correlation between enzyme-linked immunosorbent assay (ELISA) and Luminex. (**A**) Correlation between shed Syndecan-1 (SDC-1) level in pleural effusion measured by ELISA and Luminex. (**B**) Correlation between VEGF level in pleural effusion measured by ELISA and Luminex. (**C**) Correlation between Mesothelin level in pleural effusion measured by ELISA and Luminex. Spearman correlation analysis was used to assess the correlation between these two immunoassays.

**Figure 2 biomolecules-10-01113-f002:**
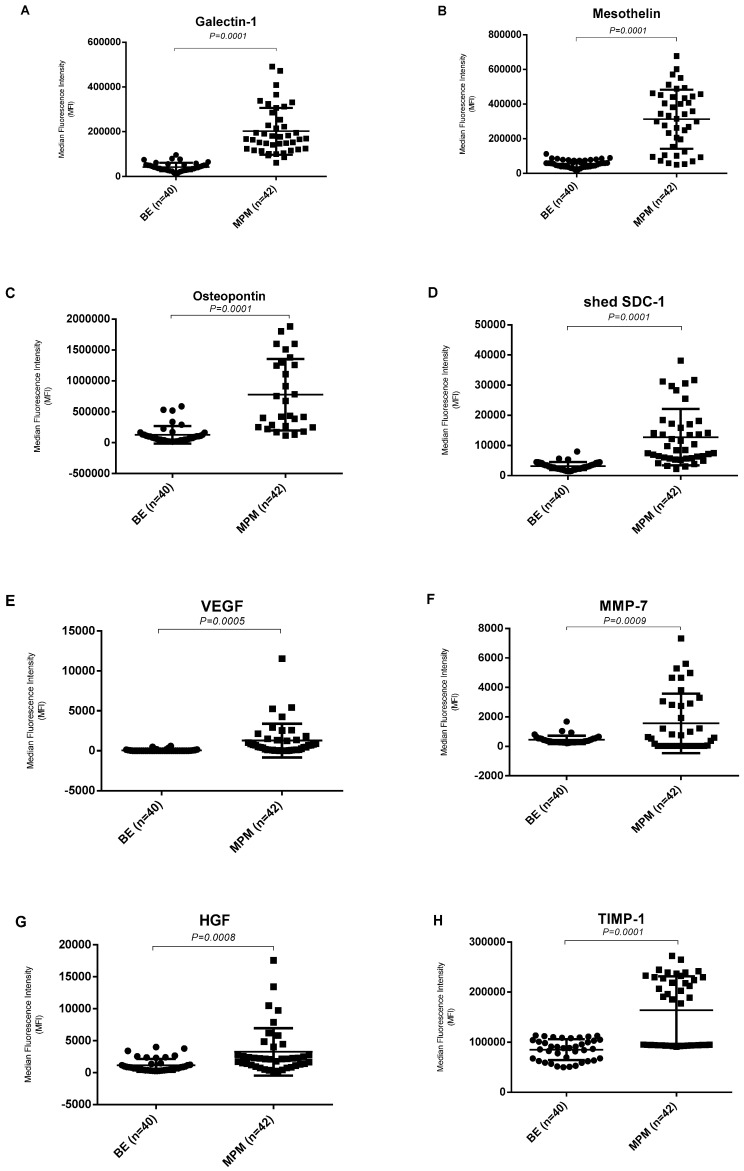
Biomarker levels in pleural effusion of malignant pleural mesothelioma (MPM) patients compared to benign effusions (BE). (**A**–**H**) Levels of Galectin-1, Mesothelin, Osteopontin, shed SDC-1, VEGF, MMP-7, HGF and TIMP-1 are significantly higher in pleural effusion from malignant pleural mesothelioma (MPM) patients (*n* = 42) comparing with benign (BE) patients (*n* = 40). (**I**) Level of NRG1-β1 is significantly lower Significance was assessed by two-tailed *t*-test at *p* ≤ 0.05.

**Figure 3 biomolecules-10-01113-f003:**
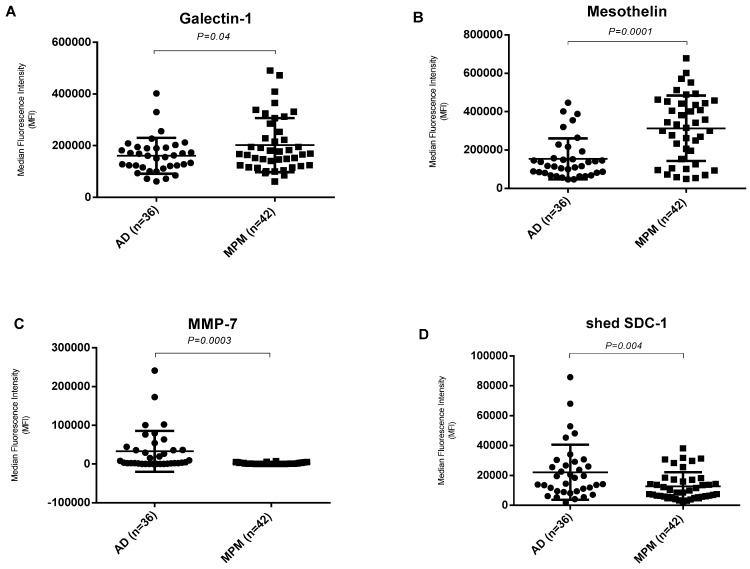
Different biomarkers for distinguishing malignant mesothelioma from metastatic adenocarcinoma patients. (**A**,**B**) Levels of Galectin-1 and Mesothelin are significantly higher in pleural effusion from malignant pleural mesothelioma (MPM) patients (*n* = 42) compared with adenocarcinoma (AD) patients (*n* = 36). (**C**,**D**) Levels of MMP-7 and shed SDC-1 are significantly lower in malignant pleural mesothelioma patients (*n* = 42) compared with adenocarcinoma patients (*n* = 36). Significance was assessed by two-tailed *t*-test at *p* ≤ 0.05.

**Figure 4 biomolecules-10-01113-f004:**
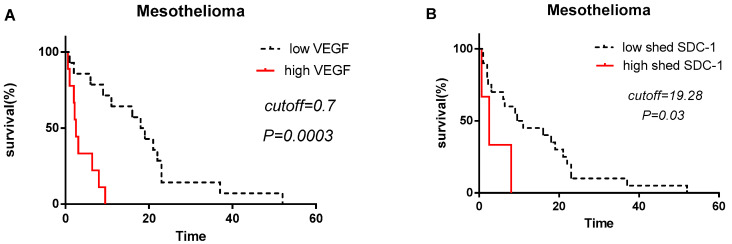
VEGF and shed SDC-1 have prognostic value in malignant mesothelioma patients. Patients were separated into “high” and “low” VEGF and shed SDC-1 level by the online web application Cutoff Finder (VEGF cutoff = 0.7 ng/mL and shed SDC-1 = 19.28 ng/mL). Malignant mesothelioma patients with high level of VEGF (**A**) and shed SDC-1 (**B**) have shorter survival time compared to malignant mesothelioma patients with low VEGF and shed SDC-1 levels. *P-*values are ≤0.05.

**Figure 5 biomolecules-10-01113-f005:**
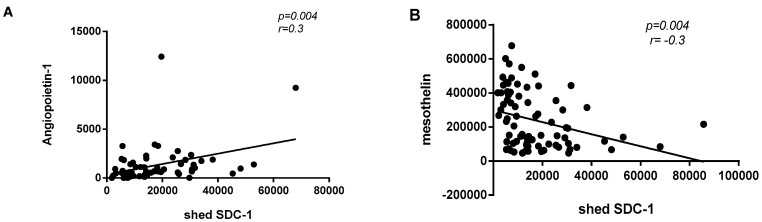
Correlation between shed SDC-1 and other biomarkers in all malignant cases. (**A**) Expression of shed SDC-1 was significantly and positively correlated with Angiopoietin-1 (*p* = 0.004 and r = 0.3), whereas, shed SDC-1 expression was negatively correlated with Mesothelin (*p* = 0.004 and r = −0.3) (**B**).

**Figure 6 biomolecules-10-01113-f006:**
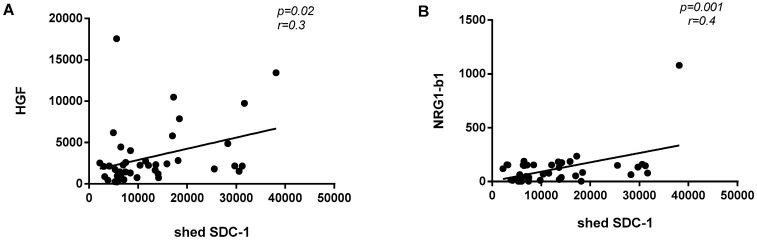
Correlation between shed SDC-1 and other biomarkers in malignant mesothelioma cases. (**A**) Expression of shed SDC-1 was significantly and positively correlated with HGF (*p* = 0.02 and r = 0.3) and NRG1-β1 (*p* = 0.001 and r = 0.4) (**B**) in malignant mesothelioma cases.

**Figure 7 biomolecules-10-01113-f007:**
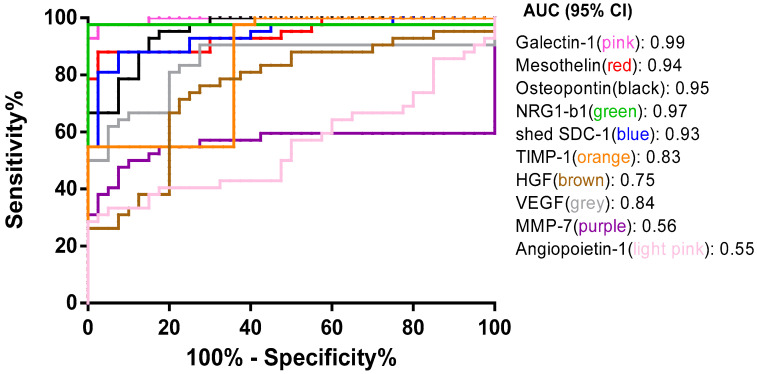
Diagnostic efficiency of individual biomarkers. Receiver operating characteristic (ROC) analysis showing the specificity and sensitivity for each individual angiogenesis related biomarkers. Area under the curve (AUC) values show that Galectin-1 (AUC = 0.99), NRG1-β1 (AUC = 0.97), Osteopontin (AUC = 0.95), Mesothelin (AUC = 0.94) and shed SDC-1(AUC = 0.93) have the most diagnostic value for malignant pleural mesothelioma.

**Table 1 biomolecules-10-01113-t001:** Parameter estimates of logistic regression model. MMP-7 has higher predictive value for distinguishing malignant pleural mesothelioma from adenocarcinoma, based on estimate and *p*-value.

Biomarkers	Estimate	Std Error	Prob > ChiSq	*p* Value
MMP-7	0.00018	0.00012	0.1586	0.00000
Mesothelin	−9.2635 × 10^−6^	2.7419 × 10^−6^	0.0007	0.00002
Osteopontin	−9.183 × 10^−7^	4.3047 × 10^−7^	0.0329	0.01

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
