# Peer review of "Multiplex Soluble Biomarker Analysis from Pleural Effusion"

_biomolecules, 2020, doi:10.3390/biom10081113_

Round 1

Reviewer 1 Report

The authors evaluated the soluble angiogenesis related molecules in pleural effusion as possible prognostic and diagnostic biomarkers for malignant pleural mesothelioma and adenocarcinomas.

I have some comments and questions:

  1. Materials and Methods:
    1. What were the primary locations of adenocarcinomas? Lungs? or there were metastatic patients with adenocarcinomas located in the diffrent organs? This  isn't clear.
    2. It is necessary to specify patients characteristics in the section: Materials and Methods
    3. Who were the patients with benign effusions? This also wasn't clear. Why did these patients have pleural effusions? The reasons are very important.
    4. Did the authors assess pleural fluid - were there any laboratory and histopathology tests of this fluid?
  2. Results:
    1. The authors should choose how to present results - only one Table 1 or Figure 2 (there are the same data)
    2. The correlation between shed SDC-1 and other biomarkers line 211; correlation between shed SDC-1 and VEGF (p=0,08 and r=0.2) is not significant,
    3. How these markers correlate with others markers of poor prognosis for example with ESR, CRP, LDH?
  3. Is it possible that these biomarkers replace histopathology and cytology in diagnosis of malignant mesothelioma
  4. References:
    1. position 3 - what is it?
    2. position 15? what is it?
    3. positions: 25, 26, 28, 29 - what is it?
    4. positions: 43, 45, 48, 50, 51, 52, 56 - what are they?

Author Response

Reviewer 1

  1. Material and Methods

1.What were the primary locations of adenocarcinoma? Lungs? Or there were metastatic patients with adenocarcinomas located in the different organs? This is not clear.

Authors response: Thank you for pointing this out. The reviewer is correct, we have added more information about adenocarcinoma patients. Metastatic adenocarcinoma cases were derived from lung-, breast-, gastroitestinal-, ovarian carcinoma and cancer of unknown primary (CUP), covering the most frequent diagnostic entities. This information added to the study design part of the material and methods on page 3 of the manuscript.

2.It is necessary to specify patients’ characteristics in the section: Materials and methods

Authors response: thank you for pointing this out. Patient characteristics are specified with regards to diagnoses in the current paper.

3.Who were the patients with benign effusions? This also was not clear. Why did these patients have pleural effusions? The reasons are very important.

Authors response: Patients with benign effusion comprised the whole spectrum of reactive mesothelial proliferations due to chronic inflammation and heart failure. The inclusion criteria were that the patient should not have any previous known cancer and should be free of cancer diagnosis also one year after the diagnosis. This information added to the study design part of the material and methods on page 3 of the manuscript.

4.Did the authors assess pleural fluid, were there any laboratory and histopathology tests of this fluid?

Authors response: All effusions went through rigorous diagnostic procedures and the diagnostic procedure followed the international Guidelines comprising morphological assessment, immunocytochemistry, FISH analysis, and in mesothelioma cases also established soluble biomarker analyses such as hyaluronan and mesothelin by ELISA.

  1. Results
  2. The authors should choose how to present results- only one table or figure.

Authors response: Thank you for pointing this out. We have made changes which the reviewer asked for it and showed the results with graphs. All the tables which showed the same results with some graphs (table 1, table 2 and table 4) have been removed from the manuscript.

  1. The correlation between shed SDC-1 and other biomarkers line 211; correlation between shed SDC-1 and VEGF (p=0,08 and r=0,2) is not significant.

Authors response: we agree that correlation between shed SDC-1 and VEGF is not significant and we apologize for this error. We have corrected figure 5, as suggested.

  1. How these markers correlate with other markers of poor prognosis for example with ESR, CRP, LDH?

Authors response: We did not assess these markers in the current study.

  1. Is it possible that these biomarkers replace histopathology and cytology in diagnosis of malignant mesothelioma?

Authors response: This is a valid and important question, and we are actively pursuing the answer. Biomarker analyses are complementary to cytology and histology, and they will not replace them.

  1. References:

Position 3-

Position 15

Position 25,26,28,29

Positions 43,45,48,50,51,52,56

Authors response: Thank you for pointing this out and we apologize for this error in the text. We have corrected all these positions in the references part of the manuscript.

Reviewer 2 Report

The manuscript titled “Multiplex soluble biomarker analysis from pleural effusionis an interesting and important study. The experimental approach is adequate. The presentation of results is clear. In this study, authors suggest that MMP-7, shed SDC-1, Mesothelin, and Galectin-1 could be diagnostic biomarkers and VEGF and SDC-1 prognostic markers in malignant pleural mesothelioma (MPM) patients. The results confirm previous findings and add new insights on the biomarkers of mesothelioma.

Minor comments;

  1. Authors should also confirm these findings in the in-vitro Authors should quantify and compare the levels of these biomarkers in the culture supernatant of normal mesothelial cells (Met5A or LP9) and MPM cell lines.
  2. It would be interesting to check the levels of HMGB1 and its correlation with these biomarkers in the pleural effusions of different groups as HMGB1 is a potential biomarker and prognostic marker of mesothelioma.

Author Response

Response to reviewer 2

Minor comments.

  1. Authors should also confirm these findings in the in-vitro Authors should quantify and compare the levels of these biomarkers in the culture supernatant of normal mesothelial cells (Met5A or LP9) and MPM cell lines.

Authors response: Thank you for pointing this out. Normal mesothelial cells are assessed in the benign patient derived samples, as they contain varying amount of reactive mesothelial cells. These cells offer the most relevant comparison. Cell culture conditions are not reliably recapitulating the benign and malignant conditions particularly regarding the representation of a mixed cell population containing, inflammatory cells, reactive mesothelial cells and in malignant cases tumor cell heterogeneity. 

  1. It would be interesting to check the levels of HMGB1 and its correlation with these biomarkers in the pleural effusions of different groups as HMGB1 is a potential biomarker and prognostic marker of mesothelioma.

Authors response: Yes, we fully agree with the reviewer on this point. It will be interesting to include new and established markers in the future to the biomarker panel. This was however beyond the scope of the current study.

Round 2

Reviewer 1 Report

Dear Authors

Thank you for your answer to my comments and your corrections of yours manuscript.

I have only one question to the authors' answer related to benign effusion. "The inclusion criteria were .....the patient should be free from cancer diagnosis also one year after the diagnosis". I understand that last word  means - the diagnosis of benign effusions.

Could the authors confirm? 

Author Response

we confirm the reviewer’s question: The patients diagnosed with benign reactive effusions were included if the they did not develop cancer the following 12 months after the diagnosis. We checked their medical records.